# Technical note: A closed chamber method to measure greenhouse gas fluxes from dry aquatic sediments

Lukas Lesmeister, Matthias Koschorreck

Department Lake Research, Helmholtz Centre for Environmental Research - UFZ, Magdeburg, 39114, Germany

5 *Correspondence to*: Matthias Koschorreck (matthias.koschorreck@ufz.de)

**Abstract.** Recent research indicates that greenhouse gas emissions from dry aquatic sediments are a relevant process in the carbon cycle of freshwaters. However, fluxes are difficult to measure because of the often rocky substrate and the dynamic nature of the habitat. Here we tested the performance of different materials to seal a closed chamber to stony ground both in laboratory and field experiments. Using on-site material consistently resulted in elevated fluxes. The artefact was caused

10 both by outgassing of the material and production of gas. The magnitude of the artefact was site dependent – the measured $CO_2$ flux was increased between 10 and 208%. Errors due to incomplete sealing proved to be more severe than errors due to non-inert sealing material.

Pottery clay as sealing material provided a tight sealing of the chamber to the ground and no production of gases was detected. With this approach it is possible to get reliable gas fluxes from hard-substrate sites without using a permanent

15 collar. Our test experiments confirmed that $CO_2$ fluxes from dry aquatic sediments are similar to $CO_2$ fluxes from terrestrial soils.

## Introduction

$CO_2$ emissions from dry freshwater systems represent a largely overlooked process in the global carbon cycle. Recent research indicates that drying and rewetting of freshwater sediments creates hot spots of carbon mineralization and thus $CO_2$

20 emissions, which are probably relevant on a global scale (Gomez-Gener et al., 2015; Reverey et al., 2016; Von Schiller et al., 2014). However, existing knowledge is scarce and mainly based on regional studies from e.g. the U.S.A. (Gallo et al., 2014), Spain (Gómez-Gener et al., 2016), the UK (Gilbert et al., 2016), or Italy (Bolpagni et al., 2017).

One reason for the shortage of available data is probably the lack of a reliable method to measure sediment-atmosphere gas fluxes in these habitats. The closed chamber approach is the most widespread method to measure gas fluxes from terrestrial

25 habitats on a small scale (Livingston and Hutchinson, 1995). The method has been extensively tested (Christiansen et al., 2011; Pumpanen et al., 2004) and is generally accepted to give good results. However, standard closed chamber protocols cannot be used in most dry aquatic systems because sealing the chamber to the ground is difficult. Even small leaks can significantly affect flux measurements (Hoffmann et al., 2017; Hutchinson and Livingston, 2001). In soil science, the chamber is often pushed into the soil to seal it towards the atmosphere. If that is not easily possible or if repeated

measurements at the same spot are planned, a permanent collar is installed. Pushing the chamber into the soil also minimizes lateral diffusion through the soil under the chamber (Hutchinson et al., 2000). This approach has been successfully used to quantify GHG fluxes from muddy dry aquatic sediments (Jin et al., 2016; Koschorreck, 2000). However, dry sediments in streams or at the shore of lentic waterbodies at low water level are often rocky and pushing the chamber into the ground is not possible. Installation of a permanent collar is also problematic, because of the dynamic nature of the habitat. Under flooded conditions, a collar will affect hydrodynamics and might change sedimentation patterns. In streams, turbulence created by a permanent installation might erode the sediment.

There are different options for sealing a chamber to the ground. The use of flexible rubber gaskets (Gilbert et al., 2016) is often not possible, because especially streambeds are often stony. Using a weighted foil (Steudler and Peterson, 1985) is also difficult in the presence of larger stones. The most promising option is the use of a ductile material. In the past, sediment material collected from the site has been used to seal the chamber (Gomez-Gener et al., 2015). However, such procedure might produce artefacts, because the sealing material is not inert and might produce especially $CO_2$. Thus, there is currently no reliable method available to measure greenhouse gas (GHG) fluxes from stony dry sediments.

In this study we conducted laboratory tests with different sealing materials to check for tightness and inertness. We then applied the most promising material in a field trial. As a result we give recommendations on how to perform GHG flux measurements in dry aquatic sediments.

## Materials and methods

### 1.1 sealing materials

We tested both commercial sealing materials (pottery clay (Töpfereibedarf Dorothea Weber, Magdeburg, Germany) and a silicon material (Silly Putty, Dow Corning, Midland, Michigan, United States) as well as different natural materials from a streambed (sand, mud) (Table 1). The natural material was collected by a shovel not reaching below a depth of approximately 5 cm and stored in polypropylene boxes until use. If not directly used for on-site measurements, it was stored under laboratory conditions and used within 3 days. To test the effect of a biologically very active material, a part of the sand was amended with glucose to stimulate $CO_2$ production.

### 1.2 test for inertness

To test, whether a material produced or consumed $CO_2$ or $CH_4$ we put 20-30 g (depending on density of material) of each material into a 1000 ml glass with a twist off lid. If possible the material was portioned into ten beads of about 1.5 cm diameter. The gas analyzer was connected with PTFE tubing to two Swagelock® connectors which were installed in the lid. The gas in the glass was then circulated through the gas analyzer and back into the glass and changes of gas mixing ratios were monitored for seven minutes. From the linear increase of the mixing ratio in the last five minutes of incubation, gas production rates were calculated. The detection limit for 25 g of material was 2.24, 0.07, and 0.02 mmol $kg^{-1}$ $d^{-1}$ for $CO^2$,

CH$_4$, and N$_2$O respectively. We performed three replicate measurements. Tests were performed between 24 and 28 °C. After each measurement at ambient conditions we lowered the CO$_2$ concentration in the glass to about 140 ppmv by flushing with argon to look for eventual outgassing of the material and then measured potential CO$_2$ increase as described before for ambient conditions.

**1.3 test for tightness**

To test the effectiveness of different materials in sealing the chamber, we used a laboratory setup. A custom made closed chamber made from opaque PVC tube, inner diameter 16 cm, height 8.06 cm with 2 Swagelock® connectors was placed on a paving slab made of exposed aggregate concrete (Figure S1). The rough surface created by the pebbles in the concrete resulted in gaps of variable shape and diameter between stone and chamber. The particular sealing material was placed around the chamber and pressed to seal the gaps. The chamber was connected to the gas analyser and the whole system flushed with Ar to lower the CO$_2$ mixing ratio to near zero. Then the Ar supply was stopped and the gas inside the chamber was circulated through the gas analyser and back. We performed 3 repetitive short term (12 min) measurements. After the third measurement we kept the chamber in place and continued recording the mixing ratio of CO$_2$ and CH$_4$ for up to 17 h.

**1.4 field test**

In order to confirm the main findings from the laboratory experiments and to test application of the sealing materials under realistic conditions, field tests at three different locations were made. Tests were carried out on stony sediments at the Elbe river and in the drawdown area of Rappbode Reservoir, a drinking water reservoir in the Harz mountains (Rinke et al., 2013). The same setup as used in the laboratory test was brought to the field (Figure S2). The chamber was placed on the ground and sealed using either clay or material from the site (Figure S3). We wetted our fingers before handling the clay to increase its plasticity. At Rappbode Reservoir we also applied a commercially available soil respiration chamber in combination with an IR-analyzer (SRC+EGM4, PP-Systems, Amesbury, U.S.). Air temperatures during field tests were between 25-27 °C at bridgesoil site, 16-19 °C at river sand site, and 27-31 °C at Rappbode reservoir respectively. Three replicate measurements were done at exactly the same site.

**1.5 analysis**

We measured the CO$_2$ concentration in 30 s intervals with a Fourier-Transform-Infrared (FTIR) Spectrometer (GASMET DX4000, Temet Instruments, Finland) after passing the gas stream through an in-line moisture trap (Drierite, USA) at a rate of 2.9 L min$^{-1}$. The standard deviation of the CO$_2$ analysis at ambient concentrations was 3 ppmv. Thus the detection limit for CO$_2$ change rates in our 5 minute laboratory incubations was 864 ppm d$^{-1}$.

For the field tests we calculated the flux of CO$_2$ [mmol m$^{-2}$ d$^{-1}$] from the linear rate of change of CO$_2$ inside the chamber:

$$J = \frac{dp_i}{dt} * F * h_{eff} * 10^{-3} \tag{1}$$

with $\frac{dp_i}{dt}$ as the change of the mixing ratio with time [ppm d$^{-1}$], F [mol m$^{-3}$] is a unit conversion factor (formula 2), and $h_{eff}$ as the effective height of the chamber headspace (formula 3). F in mol m$^{-3}$ results from

$$F = \frac{10^5 \; bar}{R * T} \tag{2}$$

with R the ideal gas constant 8.314 J K$^{-1}$ mol$^{-1}$ and T the air temperature [K]. The effective height of the chamber $h_{eff}$ = 0.12 m was calculated from the inner volume of chamber plus FTIR analyser:

$$h_{eff} = \frac{V_{chamber} + V_{GASMET}}{A} \tag{3}$$

with $V_{chamber}$ is the inner volume of the chamber, $V_{GASMET}$ the inner volume of the analyser, and A = inner surface area of the chamber. The lowest detectable $CO_2$ flux in a 5-minute measurement was 4.05 mmol m$^{-2}$ d$^{-1}$, for $CH_4$ and $N_2O$ the detection limit was 0.14 mmol m$^{-2}$ d$^{-1}$. Detection limits were calculated assuming that the concentration change during a flux measurement was equal to the minimum difference which could be measured by the analyser (3 ppm $CO_2$, 0.1 ppm $CH_4$, 0.03 ppm $N_2O$).

Together with the $CO_2$ data the FTIR analyser delivered $CH_4$ and $N_2O$ mixing ratios. Thus, we also looked for the effect of the sealing material on the fluxes of $CH_4$ and $N_2O$.

Differences between treatments were checked by a t-test after checking for normality (Shapiro-Wilk) and homogeneity of variance (Bartlett) using the software R (R-Core-Team, 2016).

**Results and discussion**

The ability of different sealing materials to provide a tight sealing was tested with the closed chamber on a concrete plate in the laboratory. These tests showed that the choice of the sealing material affected the outcome of the measurements. All sealing materials produced an initial increase of the $CO_2$ mixing ratio in the chamber if measurements started at low $CO_2$ (Figure 1a). However, only in the chambers sealed with clay or the silicon material the mixing ratio became constant at a level well below ambient, which shows that these materials provided a tight sealing. Furthermore these results show that both materials did not produce $CO_2$ (Table 1). All the on-site materials resulted in continuously rising $CO_2$ concentrations well above the ambient mixing ratio of about 400 ppmv. This clearly shows that the on-site materials produced $CO_2$ and that this potentially affected the $CO_2$ concentration in the chamber. Similar to $CO_2$ also $CH_4$ initially increased in all experiments (Figure 1b). For clay the mixing ratio levelled off at about 1 ppm, well below the atmospheric concentration. This confirms that clay provided a tight sealing also for less water soluble gasses and shows that clay did not produce $CH_4$. With the other sealing materials $CH_4$ did not reach the atmospheric concentration during the experiment except with river mud which clearly produced $CH_4$.

The inertness of the material was further tested by incubating sealing material in closed incubation vessels. These experiments confirmed that indeed the on-site material produced $CO_2$ with variable rate while $CO_2$ production by clay or putty was at or below the detection limit (Table 2). This raises the question for the reason for the initial rise in $CO_2$ in the clay and putty sealed chamber. Our chamber tests on the concrete plates showed that there was an initial increase of $CO_2$ in clay or putty sealed chambers only at artificially lowered mixing ratio (Table 2). If measurements were performed at ambient $CO_2$, $CO_2$ in the clay and putty sealed chambers remained constant, while it was increasing in the chambers sealed with the on-site materials. Outgassing of the material was also observed when incubating the sealing material in closed vials at artificially lowered $pCO_2$. We conclude that the initial rise in $CO_2$ with clay and putty was caused by out-gassing of the material rather than by production of $CO_2$. We did not detect a significant production of $CH_4$ in our inertness experiments (data not shown). However, our FTIR analyser showed some outgassing of ammonia from the putty material (data not shown). Therefore, but also because it is cheaper and environment friendly, we performed our field tests only with clay.

Field tests showed consistently higher $CO_2$ fluxes if the on-site material was used compared to clay (Figure 2). The deviation was different between sites from a small but still significant (p=0.04) difference of 10 % at the bridge soil site to up to 208 % at the sandy river site. Thus, using on site material to seal chambers produces a site depending over-estimation of the $CO_2$ flux. The results obtained with clay at the reservoir site were similar to the measurements with a tested (Pumpanen et al., 2004) soil respiration chamber, showing the reliability of our measurement setup. Compared to the artefact by non-inert sealing material, the effect of incomplete sealing was even worse. Leakage of the chamber resulted in non-linear concentration changes during measurements and very low $CO_2$ fluxes (Figure 3).

We never observed a significant flux of $CH_4$ during our field tests, confirming earlier results which showed very low fluxes of $CH_4$ from dry sediments (Gomez-Gener et al., 2015). However, if the chamber was sealed with the on-site material a small $CH_4$ flux of up to 4.1 mmol $m^{-2}$ $d^{-1}$ was detected at the reservoir site. This shows that when analysing $CH_4$ fluxes, care must be taken not to use methanogenic material to seal the chamber.

We also observed small fluxes of $N_2O$. At the bridgesoil site, the flux of 0.12 ± 0.02 mmol $m^{-2}$ $d^{-1}$ was the same with clay and on site material. In contrary, at the reservoir site using clay, the $N_2O$ flux was below the detection limit while a flux of 0.09 ± 0.04 mmol $m^{-2}$ $d^{-1}$ was measured when the on-site material was used. Thus, similar to $CH_4$, using sealing material from an anoxic zone might create an artificial $N_2O$ flux.

Not inserting the chamber into the sediment might enable lateral diffusion beneath the chamber walls. This may affect flux measurements especially at sites with porous sediments. Potential artefacts can be minimized by keeping the measuring time as short as possible – a few minutes only (Hutchinson and Livingston, 2001). Lateral diffusion is not a problem in waterlogged or compact sediments.

Sealing the chamber in the field with clay proved to be convenient. For ensuring good sealing performances using clay, wetting the sealing material directly prior to use proved to be useful to increase mouldability and to enhance adhesion to the ground and to the chamber. It is well known that wetting of dry soils triggers $CO_2$ production (Birch, 1958). In our experiment, the clay was slightly wetted but the data do not show any $CO_2$ production (Figure 1). Thus, wetting the clay to

increase its plasticity was not a problem. Forming sausage-shaped roles of clay that could be placed quickly around the chamber lead to a quick and easy sealing procedure taking 1-2 min depending on practise and nature of the surroundings (Figure S3). To prevent concentration changes in the chamber during the sealing process, we recommend the use of a chamber with removable lid. Such a chamber could be sealed to the ground while open to the atmosphere. The clay can be reused after each measurement. However, care must be taken to remove adhering soil particles. Tests had shown that using dirty clay has the potential to produce artefacts (data not shown).

The fluxes between 75 and 241 mmol $m^{-2}$ $d^{-1}$ are very similar to results obtained in Spain (209 ± 10 mmol $m^{-2}$ $d^{-1}$, (Gomez-Gener et al., 2015) and in Arizona (44 mmol $m^{-2}$ $d^{-1}$, (Gallo et al., 2014)). Our measurements in two contrasting temperate habitats confirm that dry sediments emit similar amounts of $CO_2$ as soils (Raich and Schlesinger, 1992).

**Conclusions**

When measuring with closed chambers on rocky ground the most important concern is to get a proper sealing between chamber and atmosphere. Indicators for leakage are non-linear concentration changes in the chamber and extremely low fluxes. We strongly recommend the use of an inert sealing material. Pottery clay proved to be both convenient and effective. We do not completely exclude the use of on-site material, but checks are necessary if the particular material does produce artefacts. Our study demonstrates that even at stony dry aquatic sediment sites closed chamber measurements of greenhouse gas fluxes are feasible under controlled and reproducible conditions.

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

## Tables

**Table 1: Substances tested as sealing materials.**

| substance | description |
|---|---|
| Clay | Potter's clay |
| River sand | Sand collected from the shores of the river Elbe (coordinates: 52° 7.92' N, 11° 39.42' O), sieved (2 mm) and homogenized before use. |
| Putty | "Silly Putty", dilatant compound by DOW CORNING® based on silicon polymers. |
| River sand + glucose | Sand collected from the shores of the river Elbe, spiked with a high concentration glucose solution ($100 \, g \, L^{-1}$), incubated for 2 days = positive control for biological activity. |
| River mud | Mud collected from the shores of the river Elbe (coordinates: 52° 7.62' N, 11° 39.04' O) |
| Bridgesoil | Fine particulate soil collected from fluvial deposits of the river Elbe (coordinates: 52° 7.62' N, 11° 39.04' O) |

5    **Table 2: Performance of different sealing materials in lab experiments (mean±SD). Number in brackets indicates number of replicates below the detection limit (DL); n=3 except for River mud (n = 4) and River sand + glucose (n = 6). Chamber test: chamber on paving slab, final flux in long term experiment (Figure 1). Chamber low $CO_2$: chamber on paving slab gassed with $N_2$ prior to measurement. $CO_2$ production: sealing material in glass vessel.**

| | Chamber test [mmol m$^{-2}$ d$^{-1}$] | Chamber low $CO_2$ [mmol m$^{-2}$ d$^{-1}$] | $CO_2$ production [mmol kg$^{-1}$ d$^{-1}$] |
|---|---|---|---|
| Clay | < DL | 29 ± 1 | 2.5 ± 0.7 (1) |
| Putty | < DL | 27 ± 5 | < DL |
| River mud | 5.94 | 20 ± 11 | 15.3 ± 3.2 |
| River sand | 10.80 | 42 ± 29 | 2.9 ± 3.7 |
| River sand + glucose | 55.38 | 104 ± 50 | 31.7 ± 2.4 |
| Bridgesoil | 5.94 | 22 ± 8 | 8.4 ± 1.6 |

**Figures**

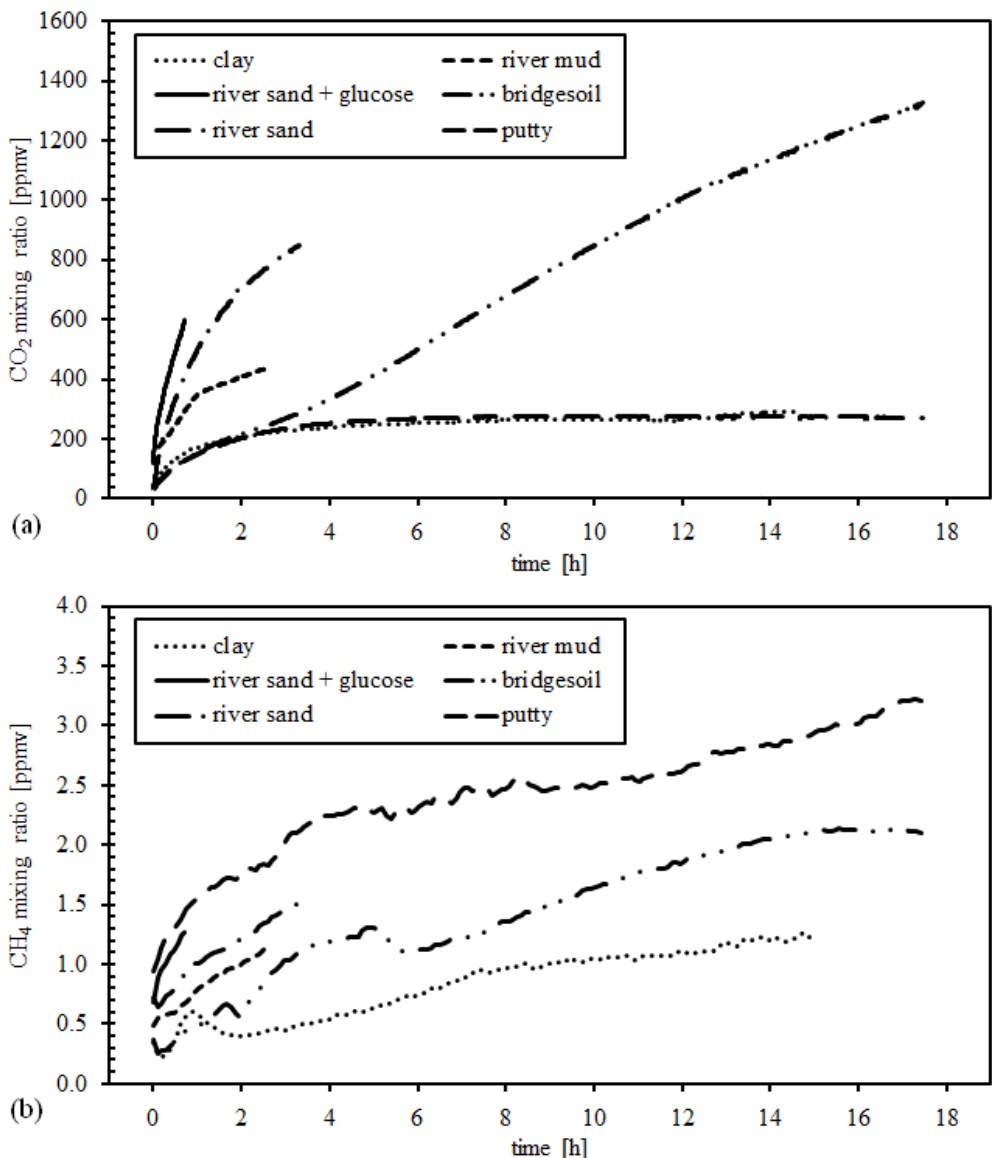

**Figure 1: CO₂ (a) and CH₄ (b) mixing ratio during long-term laboratory chamber experiments with different sealing materials.**

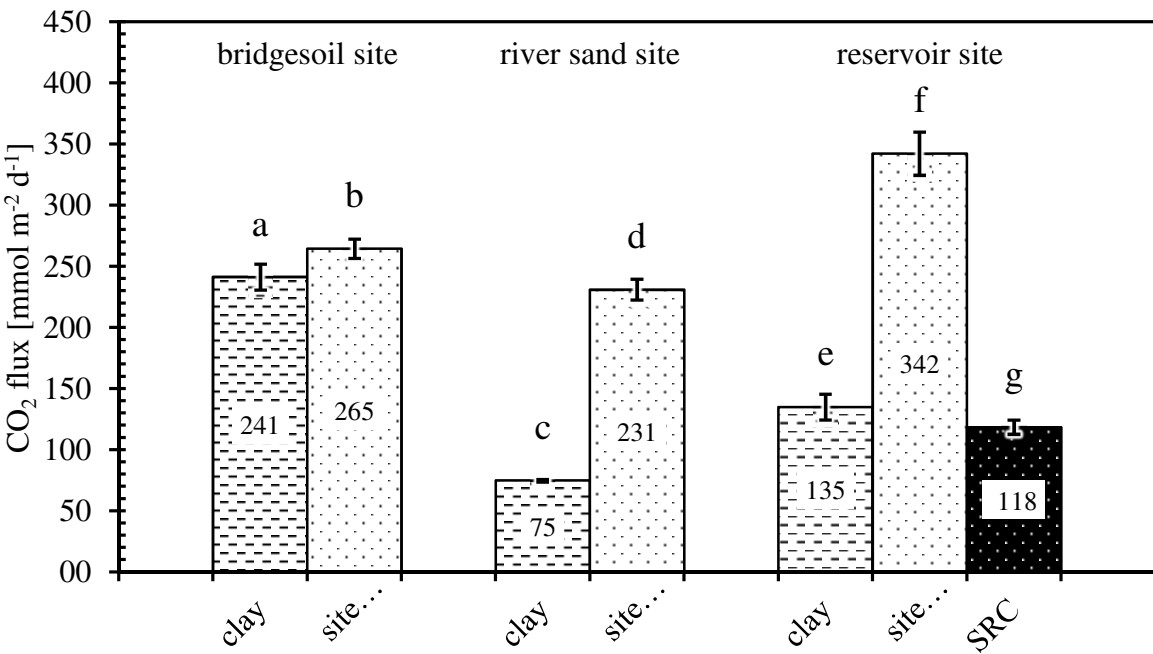

**Figure 2: Mean CO$_2$ fluxes detected in the different sample groups during field experiments (mean ± SD, n = 3; reservoir site n = 4). SRC = soil respiration chamber + IR analyser. Different letters indicate significant difference between columns.**

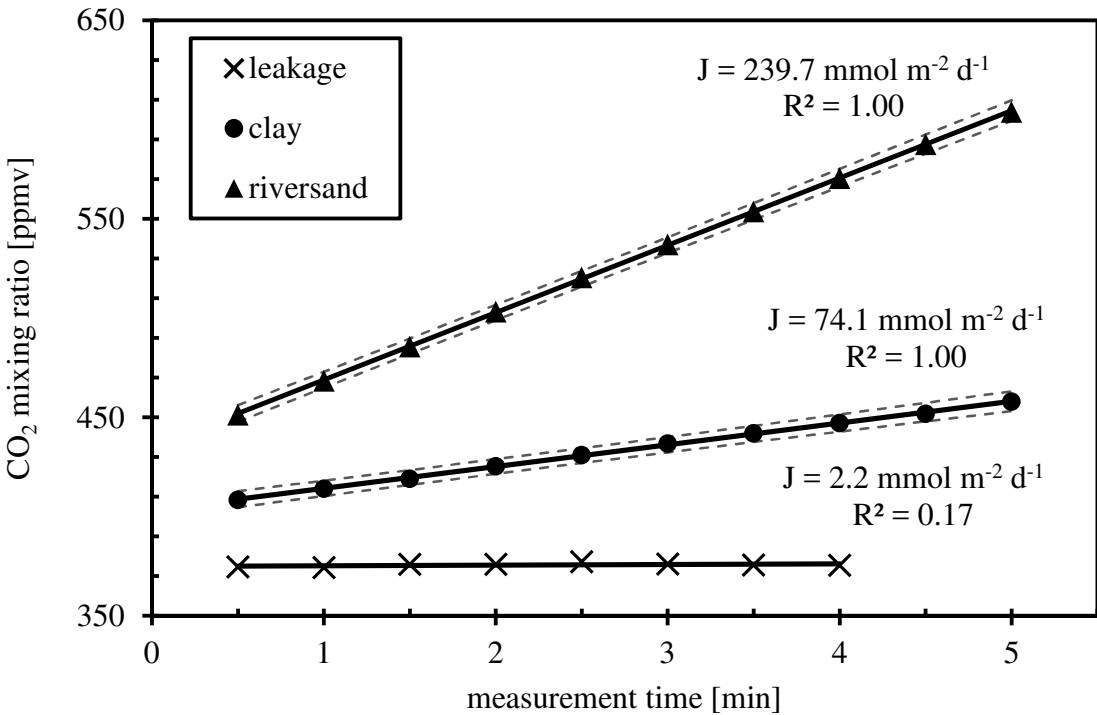

**Figure 3: Example showing typical progression of $CO_2$ concentration during field measurements with clay and ambient material (example: river sand site). For "leakage" measurements the chamber was placed on the sediment without applying any sealing material (small holes were visible). Dotted lines indicate 99.9999% error bands.**