# Peer review of "Technical note: A closed chamber method to measure greenhouse gas fluxes from dry aquatic sediments"

_Atmospheric Measurement Techniques, 2016_

## Referee Comment (RC1) · Anonymous Referee #1 · 17 Feb 2017

The technical note addresses the problem of sealing a chamber to a stony ground for performing flux measurements. Getting chambers gas-tight under such environmental conditions is indeed a problem. The note describes the testing of different sealing materials on the flux of $CO_2$ and found that potting clay was a reliable sealing material. The study addressed inertness and tightness of the sealing materials in the lab and applied the sealing techniques under field conditions.

I found this paper a nice short story on a technical problem of wide interest to people working with chamber techniques to measure gas fluxes. The study is well done and well described. I assume it will be of interest to many readers.

General comments: 1.Chamber measurement on soil ground also have the problem of

tightness and gas diffusion through the soil between inside and outside the chamber. These general problems are only much stronger when the ground consists of gravel or stones instead of soil. The authors may wish referring to older literature on diffusion through soil below the chamber walls, e.g. G. P. Livingston and G. L. Hutchinson. Enclosure-based measurement of trace gas exchange: applications and sources of error. edited by P. A. Matson and R. C. Harriss, Oxford:Blackwell, 1995, p. 14-51; or, G. L. Hutchinson, G. P. Livingston, R. W. Healy, and R. G. Striegl. Chamber measurement of surface-atmosphere trace gas exchange: Numerical evaluation of dependence on soil, interfacial layer, and source/sink properties. J.Geophys.Res. 105:8865-8875, 2000.

2.The lab tests of the sealing material were all done with $CO_2$. $CO_2$ is a water soluble gas, which may behave differently than other atmospheric gases that are not well soluble, like $H_2$, $CH_4$, CO. Some gases may also undergo chemical reactions, e.g. CO, NO, sulfur compounds. The field test addressed $CH_4$ and $N_2O$ in addition to $CO_2$. However, the tests for inertness were not done with gases other than $CO_2$. I think this problem should be addressed in the discussion.

3. The data shown in the bar graph (Fig.2) should be tested for statistically significant difference.

Technical correction: 4. L.7: the dynamic nature of the habitat is not subject of the study.

5. L.14: give the companies which supplied the materials.

---

## Referee Comment (RC3) · M. Hoffmann (Referee) · 2 Apr 2017

The study of Lesmeister and Koschorreck addresses the important issue of an airtight, non-influencing sealing strategy of chambers when measuring GHG emissions from dried aquatic sediments with coarse particles. Therefore different materials are tested within an approach, which combines a laboratory experimental setup with a short field study to find the most appropriate sealing material. In general the study is well designed and written, and thus suitable for publication in AMT. However, I have some major concerns: - within the abstract/ introduction especially the problem of coarse (rocky) material is mentioned (L7), however, the field study seems to only test the different sealing materials on sand and mud but not on coarse material (e.g. gravel). –

did you tested whether only slightly inserting the chamber into the sand or mud would have yield in similar results during the field study (important for generalization of made statements!)? - testing silicone for sealing is mentioned in section 1.1 (L14), but not tested during the field study (or mentioned) - was the wetted clay tested during the laboratory test as well (L14: "a little water was added to the clay")? - I am curious about the drastic differences between on-site material and clay used for sealing at the river sand site and at the reservoir site during the field study. From where was the on-site material taken to seal the chamber? Might it be that the measurements were generally disturbed cause the material was taken from around the chamber? - is it right, that the field study only consisted of three to four repetitious measurements per sealing material and site? - might the time needed for sealing (1-2 min) yield in an already increased chamber starting concentration which biases lateron flux calculation? Depite of this, there are also some minor concerns: - Did you test for saturation effects (due to small chamber size and rather high $CO_2$-emissions)? - what kind of statistic test was performed (P4, L24) and was the test performed for the n of only 3-4 - in general more details about used statistics are needed! Statistical tests comparing the fluxes should be added to this figure. The low n should be mentioned here. - P1 L4 erases "probably" - P7 Tab. 2 caption: capitalize "number" - Fig. 1: check the y-axis? Why was the incubation time different for the different materials? Could you add a error band around the line displaying the deviation during the three repetitive measurements (same for Fig. 3)? - Fig. 3: how was the leakage measurement performed during the field study? - please add "aquatic" to the titel ("dry aquatic sediments") - How does Lorke et al. fit as a reference to the MS, if measurements were not performed on dried sediments but water (floating chamber)? - Numbering is wrong (1.3 comes before 1.2) - please add a space between 28 and °C at P2 L26 - please directly address that the laboratory test is only able to detect the combined effect of leakage and $CO_2$ production (which is still suitable for the purpose of the study)!

---

## Author Response (AR1)

**Technical note: A closed chamber method to measure greenhouse gas fluxes from dry aquatic sediments Response to the reviews**

5    We appreciate the fair reviews and address all points raised by the reviewers in our detailed response below. Our replies to the reviews are in red letters as well as any changes in the manuscript.

**Reply to RC1**

The technical note addresses the problem of sealing a chamber to a stony ground for performing flux measurements. Getting chambers gas-tight under such environmental conditions is indeed a problem.

10    The note describes the testing of different sealing materials on the flux of CO2 and found that potting clay was a reliable sealing material. The study addressed inertness and tightness of the sealing materials in the lab and applied the sealing techniques under field conditions. I found this paper a nice short story on a technical problem of wide interest to people working with chamber techniques to measure gas fluxes. The study is well done and well described. I assume it will be of interest to many readers.

15    General comments: 1.Chamber measurement on soil ground also have the problem of tightness and gas diffusion through the soil between inside and outside the chamber. These general problems are only much stronger when the ground consists of gravel or stones instead of soil. The authors may wish referring to older literature on diffusion through soil below the chamber walls, e.g. G. P. Livingston and G. L. Hutchinson. Enclosure-based measurement of trace gas exchange: applications and sources of

20    error. edited by P. A. Matson and R. C. Harriss, Oxford:Blackwell, 1995, p. 14-51; or, G. L. Hutchinson, G. P. Livingston, R. W. Healy, and R. G. Striegl. Chamber measurement of surface-atmosphere trace gas exchange: Numerical evaluation of dependence on soil, interfacial layer, and source/sink properties. J.Geophys.Res. 105:8865-8875,

2000. We added some references dealing with methodological aspects of chamber measurements on

25    soils – especially to the second paragraph. The Livingston and Hutchinson 1995 was already cited – but originally we did cite the whole book (Matson and Harriss), not just the chapter. We now cite the chapter. We also added a paragraph about lateral diffusion to the discussion.

2.The lab tests of the sealing material were all done with CO2. CO2 is a water soluble gas, which may behave differently than other atmospheric gases that are not well soluble, like H2, CH4, CO. Our idea

30    was to proof tightness, it is sufficient to proof that with one gas. We can assume that if the chamber is tight for one gas, the same should be true for other gases. $H_2$ diffuses faster than the other gases, but we do not think that diffusion is fast enough to affect our measurements. The solubility of $CO_2$ affects the link between $CO_2$ production (by microbial processes) and the $CO_2$ flux. The relation between $CO_2$ production and emission, however, is not part of our study.

35    To follow the reviewers idea and to include a less soluble gas, we added the $CH_4$ data to Figure 1 and discuss them in the text: "Similar to $CO_2$ also $CH_4$ initially increased in all experiments (Figure 1b). For clay the mixing ratio levelled off at about 1 ppm, well below the atmospheric concentration. This confirms that clay provided a tight sealing also for less water soluble gasses and shows that clay did not produce $CH_4$. With the other sealing materials $CH_4$ did not reach the atmospheric concentration during

40    the experiment except with river mud which clearly produced $CH_4$."

Some gases may also undergo chemical reactions, e.g. CO, NO, sulfur compounds. This is true, but our study focuses on greenhouse gases.

The field test addressed CH4 and N2O in addition to CO2. However, the tests for inertness were not done with gases other than CO2. I think this problem should be addressed in the discussion. We actually

45    measured $CH_4$ in our laboratory experiments. We added the methane data to Figure 1 and discuss them in the text as explained above. In our inertness experiments we did not detect a significant production of $CH_4$. We added this information to the text: "We did not detect a significant production of $CH_4$ in our inertness experiments (data not shown)".

3. The data shown in the bar graph (Fig.2) should be tested for statistically significant difference. Difference was checked by a t-test after checking for normality and homogeneity of variance. We add this information to the manuscript and also indicate it in the figure.

Technical correction:

4. L.7: the dynamic nature of the habitat is not subject of the study. Yes – that is right. But it is a reason why permanent collars cannot be used.

5. L.14: give the companies which supplied the materials. We added the companies.

**Reply to RC2**

The study by Lesmeister & Koschorreck addresses the problem of measuring green house gas (GHG) gas (primarily CO2) fluxes from dry aquatic sediments with coarse particles. They address this methodological issue by combining in a concise way both laboratory and field tests. My major concerns are: - The lack of consistent testing of all three GHG analyzed here (CO2, CH4, N2O). - The lack of testing of the wetting of clay. - The lack of references to studies in terrestrial soils that have addressed some of these methodological problems in the past. Also, address how the results presented here could be applied to terrestrial soils. This would make the better also more interesting for a wider audience. See also some specific comments:

P1, Title: I suggest adding "aquatic" before "dry sediments". This is a good suggestion – we changed the title.

P1, L14: I suggest using "terrestrial" instead of "normal". OK – we changed the text.

P1, L21: There are some recent studies on GHG fluxes from dry sediments from other regions too (e.g. Bolpagni, Rossano, et al. "Role of ephemeral vegetation of emerging river bottoms in modulating CO2 exchanges across a temperate large lowland river stretch." Aquatic Sciences: 1-10; Jin, Hyojin, et al. "Enhanced greenhouse gas emission from exposed sediments along a hydroelectric reservoir during an extreme drought event." Environmental Research Letters 11.12 (2016): 124003; Gilbert, Peter J., et al. "Quantifying rapid spatial and temporal variations of CO2 fluxes from small, lowland freshwater ponds." Hydrobiologia (2016): 1-11.). We added those references

P1, L22-30: Make clear that it is possible to measure GHG fluxes from aquatic sediments, but that this measures have so far been limited to fine sediments because of methodological constraints. Thanks for this suggestion. We added: "This approach has been successfully used to quantify GHG fluxes from muddy dry aquatic sediments (Hyojin et al., 2016; Koschorreck, 2000). However, dry sediments in streams or at the shore of lentic waterbodies at low water level are often rocky and pushing the chamber into the ground is not possible.".

P1, L23: "widespread". corrected

P2, L8: There is some methods, but only for fine sediments. Correct – we added this aspect as explained above and by adding "stony" at this point.

P2, L10: I think you should add "on" before "how". Corrected.

P2, L20: It is unclear if you really test CH4 flux (and N2O). We tested both $CO_2$ and $CH_4$. Some information on $CH_4$ was added to the results section: "We did not detect a significant production of $CH_4$ in our inertness experiments (data not shown).".

P2, L25: This detection limit is for CO2 and CH4? For $CO_2$. We added the detection limits for the other gases to the method section.

P2, L25: "Three replicate measurements"? Yes – corrected.

P3, L13; I miss more information on the characteristics of the chambers used. We used exactly the same chamber as in the laboratory experiments. The design of the chamber is explained on page 3, l.2-3.

P3, L14: The effect of adding water was not tested in the lab, was it? This my have influenced the results and needs at least some discussion. Unfortunately we did not perform wetting experiments with the clay. However, the clay we used was not really dry and we only added very little water to increase plasticity. In fact, we only wetted our fingers before placing the clay around the chamber. The results in Figure 1a show that the clay was not producing $CO_2$. Thus, we think that wetting of the clay did not affect our measurements. We changed in the method section: "We wetted our fingers before handling the clay to increase its plasticity". We also added to the discussion: "It is well known that wetting of dry soils triggers $CO_2$ production (Birch, 1958). In our experiment, the clay was slightly wetted but the data do not show any $CO_2$ production. Thus, wetting the clay to increase its plasticity was not a problem.".

P3, L17: Specify if the temperatures reported here and in other parts of the text are air or sediment temperatures. We measured air temperature near to the soil. This is now specified in the text.

P4, L5: It seems strange that CH4 and N2O are presented so late. The title is about GHG but then the manuscript deals mostly with CO2. What were the limits of detection for CH4 and N2O? We added the detection limits for the $CH_4$ and $N_2O$ flux: The lowest detectable $CO_2$ flux in a 5-minute measurement was 4.05 mmol $m^{-2}$ $d^{-1}$, for $CH_4$ and $N_2O$ the detection limit was 0.14 mmol $m^{-2}$ $d^{-1}$

P4, L8-L22: For clarity and consistency, the text here could refer more explicitly to the concepts of inertness and tightness.

We re-formulated the paragraph to make those concepts more clear.

Table 1: Any brand name for the clay? We do not have a brand name. It was ordinary pottery clay. We add the company were we bought it.

Figure 1: Is "CO2 mixing ratio" the correct name for the y-axis? Yes. We corrected the figure legend accordingly. Why was the incubation for some materials shorter (<4h)? As soon as the $CO_2$ mixing ratio in the chamber exceeded the atmospheric mixing ratio, it was clear that the sealing material was producing $CO2$. There was no need to continue the experiment beyond this point. That is why we stopped the experiments as soon as the atmospheric mixing ratio was exceeded. In the cases were $CO_2$ did not reach the atmospheric mixing ratio we extended the experiment to see, whether there was a slow leaking in of atmospheric $CO2$.

Figure 2: Put the units of flux in parentheses. corrected. Statistical tests comparing the fluxes could be added to this figure. Difference was checked by a t-test after checking for normality and homogeneity of variance. We added this information to the method section. We also added the information about statistical difference to the figure.

The SRC results should be highlighted more in the text. We added a comment on the SRC results in the text: "The results obtained with clay at the reservoir site were similar to the measurements with a tested (Pumpanen et al., 2004) soil respiration chamber, showing the reliability of our measurement setup."

**Reply to RC3**

The study of Lesmeister and Koschorreck addresses the important issue of an airtight, non-influencing sealing strategy of chambers when measuring GHG emissions from

5  dried aquatic sediments with coarse particles. Therefore different materials are tested within an approach, which combines a laboratory experimental setup with a short field study to find the most appropriate sealing material. In general the study is well designed and written, and thus suitable for publication in AMT. However, I have some major concerns:

10  - within the abstract/ introduction especially the problem of coarse
(rocky) material is mentioned (L7), however, the field study seems to only test the different sealing materials on sand and mud but not on coarse material (e.g. gravel). Actually, our field measurements were not done on sand or mud but on quite stony ground with a lot of gravel. The '"on-site" material used for sealing was taken in the vicinity of our measuring site. We were especially

15  looking for stone free material for sealing. Our original method description was misleading. We changed it to: "Tests were carried out on stony sediments…". Here is a photograph showing the gravel at the river site:

[Figure]

- did you tested whether only slightly inserting the chamber into the sand or mud would

20  have yield in similar results during the field study (important for generalization of made statements!)? It was not possible to insert our chamber into the substrate. At the reservoir site it was possible to insert a commercially available soil respiration chamber (with sharp edges) a little bit into the substrate. We comment on those measurements in the results and discussion.
- testing silicone for sealing is mentioned in section 1.1 (L14), but not tested during the field study (or

25  mentioned) Yes – that is right. Both clay and putty performed acceptable in the laboratory test. We decided to proceed only with clay, because it was cheaper and environment friendly. Furthermore the putty material was releasing ammonia. We added this information to the results.
- was the wetted clay tested during the laboratory test as well (L14: "a little water was added to the clay")? Unfortunately we did not perfom an "inertness experiment" with wetted clay. As explained in

30  our reply to reviewer 2 we do not think that wetting the clay triggered microbial activity. We now address this aspect in the discussion.
- I am curious about the drastic differences between on-site material and clay used for sealing at the river sand site and at the reservoir site during the field study. From where was the on-site material taken to seal the chamber? Might it be that the measurements were generally disturbed cause the material was

35  taken from around the chamber? No – the material was taken several meters away from the chamber.
- is it right, that the field study only consisted of three to four repetitious measurements per sealing material and site? Yes. The effects were quite clear. So we decided that more replicates were not necessary.
- might the time needed for sealing (1-2 min) yield in an already increased chamber starting

40  concentration which biases lateron flux calculation? Yes – there was slight increase during sealing – but only a few ppm of $CO_2$. ppm versus time curves were apparently linear. We do not think that this affected our measurements. For future measurements, however, we recommend the use of a chamber with removable lid. Then the sealing can be done while the chamber is still open to the atmosphere. We added this suggestion to the discussion: "To prevent concentration changes in the chamber during the

sealing process, we recommend the use of a chamber with removable lid. Such a chamber could be sealed to the ground while open to the atmosphere."

Depite of this, there are also some minor concerns:

- Did you test for saturation effects (due to small chamber size and rather high CO2-emissions)? We are not sure whether we understand this comment. We did not perform long term measurements in the field. We wanted to measure *in-situ* fluxes. Thus, we tried to do the measurements as fast as possible.

- what kind of statistic test was performed (P4, L24) and was the test performed for the n of only 3-4. - in general more details about used statistics are needed! Statistical tests comparing the fluxes should be added to this figure. The low n should be mentioned here. Difference was checked by a t-test checking for normality and homogeneity of variance. We add to the methods: "Differences between treatments were checked by a t-test after checking for normality (Shapiro-Wilk) and homogeneity of variance (Bartlett) using the software R (R-Core-Team, 2016)."

- P1 L4 erases "probably" The reviewer is probably right, but we think that there are currently not enough data to be 100% sure about this statement. There is actually a global initiative trying to verify the global relevance (http://www.ufz.de/dryflux/). We would like to keep the "probably"

- P7 Tab. 2 caption: capitalize "number" corrected.

- Fig. 1: check the y-axis? The Y-axis label is correct. We changed the figure legend accordingly. Why was the incubation time different for the different materials As soon as the $CO_2$ mixing ratio in the chamber exceeded the atmospheric mixing ratio, it was clear that the sealing material was producing $CO_2$. There was no need to continue the experiment beyond this point. That is why we stopped the experiments as soon as the atmospheric mixing ratio was exceeded. In the cases were $CO_2$ did not reach the atmospheric mixing ratio we extended the experiment to see, whether there was a slow leaking in of atmospheric $CO_2$.

Could you add a error band around the line displaying the deviation during the three repetitive measurements (same for Fig. 3)? Figure 1 shows the results of single measurements. Thus, it is not possible to ad error bands. Obviously, our method description was not clear here. We modified it to: "We performed 3 repetitive short term (12 min) measurements. After the third measurement we kept the chamber in place and continued recording the mixing ratio of $CO_2$ and $CH_4$ for up to 17 h."

We added error bands to Figure 3.

- Fig. 3: how was the leakage measurement performed during the field study? We just placed the chamber on the sediment without any sealing. There were small holes visible. We extended the figure legend accordingly.

- please add "aquatic" to the titel ("dry aquatic sediments") OK – we changed the title.

- How does Lorke et al. fit as a reference to the MS, if measurements were not performed on dried sediments but water (floating chamber)? We cited that paper not because it contains measurements on dry sites but because it contains an innovative chamber sealing strategy: a foil sealing to a non-flat and moving water surface. We think that at least considering the use of a flexible foil fits well to our introduction. During the revision process we discovered references were a foil was used on soils. Thus, we replaced the Lorke et al reference by Steudler and Peterson (1985).

- Numbering is wrong (1.3 comes before 1.2) Right – corrected.

- please add a space between 28 and ∘C at P2 L26 corrected.

- please directly address that the laboratory test is only able to detect the combined effect of leakage and CO2 production (which is still suitable for the purpose of the study)! That is only partly right. In the cases were we had $CO_2$ production we indeed could not get information about tightness. However, in cases where we have a long term value below the atmospheric mixing ratio we are pretty sure, that the material is both inert and tight. Theoretically, a continuous leakage of $CO_2$ into the chamber could be exactly compensated by $CO_2$ consumption in the chamber. But we have no reason to assume that our chamber or sealing material consume $CO_2$ to keep the $CO_2$ concentration sustainable below ambient.

[revised manuscript text omitted]